# Achieve the Minimum Width of Neural Networks for Universal Approximation

**Yongqiang Cai**[*]
Beijing Normal University
`caiyq.math@bnu.edu.cn`

## Abstract

The universal approximation property (UAP) of neural networks is fundamental for deep learning, and it is well known that wide neural networks are universal approximators of continuous functions within both the $L^p$ norm and the continuous/uniform norm. However, the exact minimum width, $w_{\min}$, for the UAP has not been studied thoroughly. Recently, using a decoder-memorizer-encoder scheme, Park et al. (2021) found that $w_{\min} = \max(d_x + 1, d_y)$ for both the $L^p$-UAP of ReLU networks and the $C$-UAP of ReLU+STEP networks, where $d_x, d_y$ are the input and output dimensions, respectively. In this paper, we consider neural networks with an arbitrary set of activation functions. We prove that both $C$-UAP and $L^p$-UAP for functions on compact domains share a universal lower bound of the minimal width; that is, $w^*_{\min} = \max(d_x, d_y)$. In particular, the critical width, $w^*_{\min}$, for $L^p$-UAP can be achieved by leaky-ReLU networks, provided that the input or output dimension is larger than one. Our construction is based on the approximation power of neural ordinary differential equations and the ability to approximate flow maps by neural networks. The nonmonotone or discontinuous activation functions case and the one-dimensional case are also discussed.

## 1 Introduction

The study of the universal approximation property (UAP) of neural networks is fundamental for deep learning and has a long history. Early studies, such as Cybenkot (1989); Hornik et al. (1989); Leshno et al. (1993), proved that wide neural networks (even shallow ones) are universal approximators for continuous functions within both the $L^p$ norm ($1 \leq p < \infty$) and the continuous/uniform norm. Further research, such as Telgarsky (2016), indicated that increasing the depth can improve the expression power of neural networks. If the budget number of the neuron is fixed, the deeper neural networks have better expression power Yarotsky & Zhevnerchuk (2020); Shen et al. (2022). However, this pattern does not hold if the width is below a critical threshold $w_{\min}$. Lu et al. (2017) first showed that the ReLU networks have the UAP for $L^1$ functions from $\mathbb{R}^{d_x}$ to $\mathbb{R}$ if the width is larger than $d_x + 4$, and the UAP disappears if the width is less than $d_x$. Further research, Hanin & Sellke (2017); Kidger & Lyons (2020); Park et al. (2021), improved the minimum width bound for ReLU networks. Particularly, Park et al. (2021) revealed that the minimum width is $w_{\min} = \max(d_x + 1, d_y)$ for the $L^p(\mathbb{R}^{d_x}, \mathbb{R}^{d_y})$ UAP of ReLU networks and for the $C(\mathcal{K}, \mathbb{R}^{d_y})$ UAP of ReLU+STEP networks, where $\mathcal{K}$ is a compact domain in $\mathbb{R}^{d_x}$.

For general activation functions, the exact minimum width $w_{\min}$ for UAP is less studied. Johnson (2019) consider uniformly continuous activation functions that can be approximated by a sequence of one-to-one functions and give a lower bound $w_{\min} \geq d_x + 1$ for $C$-UAP (means UAP for $C(\mathcal{K}, \mathbb{R}^{d_y})$). Kidger & Lyons (2020) consider continuous nonpolynomial activation functions and give an upper bound $w_{\min} \leq d_x + d_y + 1$ for $C$-UAP. Park et al. (2021) improved the bound for $L^p$-UAP (means UAP for $L^p(\mathcal{K}, \mathbb{R}^{d_y})$) to $w_{\min} \leq \max(d_x + $

---
[*]School of Mathematical Sciences, Laboratory of Mathematics and Complex Systems, MOE, Beijing Normal University, 100875 Beijing, China

$2, d_y + 1$). A summary of known upper/lower bounds on minimum width for the UAP can be found in Park et al. (2021).

In this paper, we consider neural networks having the UAP with arbitrary activation functions. We give a universal lower bound, $w_{\min} \geq w^*_{\min} = \max(d_x, d_y)$, to approximate functions from a compact domain $\mathcal{K} \subset \mathbb{R}^{d_x}$ to $\mathbb{R}^{d_y}$ in the $L^p$ norm or continuous norm. Furthermore, we show that the critical width $w^*_{\min}$ can be achieved by many neural networks, as listed in Table 1. Surprisingly, the leaky-ReLU networks achieve the critical width for the $L^p$-UAP provided that the input or output dimension is larger than one. This result relies on a novel construction scheme proposed in this paper based on the approximation power of neural ordinary differential equations (ODEs) and the ability to approximate flow maps by neural networks.

Table 1: Summary of the known minimum width of feed-forward neural networks that have the universal approximation property.

| Functions | Activation | Minimum width | References |
|---|---|---|---|
| $C(\mathcal{K}, \mathbb{R})$ | ReLU | $w_{\min} = d_x + 1$ | Hanin & Sellke (2017) |
| $L^p(\mathbb{R}^{d_x}, \mathbb{R}^{d_y})$ | ReLU | $w_{\min} = \max(d_x + 1, d_y)$ | Park et al. (2021) |
| $C([0,1], \mathbb{R}^2)$ | ReLU | $w_{\min} = 3 = \max(d_x, d_y) + 1$ | Park et al. (2021) |
| $C(\mathcal{K}, \mathbb{R}^{d_y})$ | ReLU+STEP | $w_{\min} = \max(d_x + 1, d_y)$ | Park et al. (2021) |
| $L^p(\mathcal{K}, \mathbb{R}^{d_y})$ | Conti. nonpoly‡ | $w_{\min} \leq \max(d_x + 2, d_y + 1)$ | Park et al. (2021) |
| $L^p(\mathcal{K}, \mathbb{R}^{d_y})$ | Arbitrary | $w_{\min} \geq \max(d_x, d_y) =: w^*_{\min}$ | **Ours** (Lemma 1) |
| | Leaky-ReLU | $w_{\min} = \max(d_x, d_y, 2)$ | **Ours** (Theorem 2) |
| | Leaky-ReLU+ABS | $w_{\min} = \max(d_x, d_y)$ | **Ours** (Theorem 3) |
| $C(\mathcal{K}, \mathbb{R}^{d_y})$ | Arbitrary | $w_{\min} \geq \max(d_x, d_y) =: w^*_{\min}$ | **Ours** (Lemma 1) |
| | ReLU+FLOOR | $w_{\min} = \max(d_x, d_y, 2)$ | **Ours** (Lemma 4) |
| | UOE†+FLOOR | $w_{\min} = \max(d_x, d_y)$ | **Ours** (Corollary 6) |
| $C([0,1], \mathbb{R}^{d_y})$ | UOE† | $w_{\min} = d_y$ | **Ours** (Theorem 5) |

‡ Continuous nonpolynomial $\rho$ that is continuously differentiable at some $z$ with $\rho'(z) \neq 0$.  .
† UOE means the function having *universal ordering of extrema*, see Definition 7  .

## 1.1 Contributions

1) Obtained the universal lower bound of width $w^*_{\min}$ for feed-forward neural networks (FNNs) that have universal approximation properties.

2) Achieved the critical width $w^*_{\min}$ by leaky-ReLU+ABS networks and UOE+FLOOR networks. (UOE is a continuous function which has *universal ordering of extrema*. It is introduced to handle $C$-UAP for one-dimensional functions. See Definition 7.)

3) Proposed a novel construction scheme from a differential geometry perspective that could deepen our understanding of UAP through topology theory.

## 1.2 Related work

To obtain the exact minimum width, one must verify the lower and upper bounds. Generally, the upper bounds are obtained by construction, while the lower bounds are obtained by counterexamples.

**Lower bounds.** For ReLU networks, Lu et al. (2017) utilized the disadvantage brought by the insufficient size of the dimensions and proved a lower bound $w_{\min} \geq d_x$ for $L^1$-UAP; Hanin & Sellke (2017) considered the compactness of the level set and proved a lower bound $w_{\min} \geq d_x + 1$ for $C$-UAP. For monotone activation functions or its variants, Johnson (2019) noticed that functions represented by networks with width $d_x$ have unbounded level sets, and Beise & Da Cruz (2020) noticed that such functions on a compact domain $\mathcal{K}$ take their maximum value on the boundary $\partial \mathcal{K}$. These properties allow one to construct counterexamples and give a lower bound $w_{\min} \geq d_x + 1$ for $C$-UAP. For general activation

functions, Park et al. (2021) used the volume of simplex in the output space and gave a lower bound $w_{\min} \geq d_y$ for either $L^p$-UAP or $C$-UAP. Our universal lower bound, $w_{\min} \geq \max(d_x, d_y)$, is based on the insufficient size of the dimensions for both the input and output space, which combines the ideas from these references above.

**Upper bounds.** For ReLU networks, Lu et al. (2017) explicitly constructed a width-$(d_x+4)$ network by concatenating a series of blocks so that the whole network can be approximated by scale functions in $L^1(\mathbb{R}^{d_x}, \mathbb{R})$ to any given accuracy. Hanin & Sellke (2017); Hanin (2019) constructed a width-$(d_x + d_y)$ network using the max-min string approach to achieve $C$-UAP for functions on compact domains; Park et al. (2021) proposed an encoder-memorizer-decoder scheme that achieves the optimal bounds $w_{\min} = \max(d_x + 1, d_y)$ of the UAP for $L^p(\mathbb{R}^{d_x}, \mathbb{R}^{d_y})$. For general activation functions, Kidger & Lyons (2020) proposed a register model construction that gives an upper bound $w_{\min} \leq d_x + d_y + 1$ for $C$-UAP. Based on this result, Park et al. (2021) improved the upper bound to $w_{\min} \leq \max(d_x + 2, d_y + 1)$ for $L^p$-UAP. In this paper, we adopt the encoder-memorizer-decoder scheme to calculate the universal critical width for $C$-UAP by ReLU+FLOOR activation functions. However, the floor function is discontinuous. For $L^p$-UAP, we reach the critical width by leaky-ReLU, which is a continuous network using a novel scheme based on the approximation power of neural ODEs.

**ResNet and neural ODEs.** Although our original aim is the UAP for feed-forward neural networks, our construction is related to the neural ODEs and residual networks (ResNet, He et al. (2016)), which include skipping connections. Many studies, such as E (2017); Lu et al. (2018); Chen et al. (2018), have emphasized that ResNet can be regarded as the Euler discretization of neural ODEs. The approximation power of ResNet and neural ODEs have also been examined by researchers. To list a few, Li et al. (2022) gave a sufficient condition that covers most networks in practice so that the neural ODE/dynamic systems (without extra dimensions) process $L^p$-UAP for continuous functions, provided that the spatial dimension is larger than one; Ruiz-Balet & Zuazua (2021) obtained similar results focused on the case of one-hidden layer fields. Tabuada & Gharesifard (2020) obtained the $C$-UAP for monotone functions, and for continuous functions it was obtained by adding one extra spatial dimension. Recently, Duan et al. (2022) noticed that the FNN could also be a discretization of neural ODEs, which motivates us to construct networks achieving the critical width by inheriting the approximation power of neural ODEs. For the excluded dimension one, we design an approximation scheme with leaky-ReLU+ABS and UOE activation functions.

### 1.3 ORGANIZATION

We formally state the main results and necessary notations in Section 2. The proof ideas are given in Section 3 4, and 5. In Section 3, we consider the case where $N = d_x = d_y = 1$, which is basic for the high-dimensional cases. The construction is based on the properties of monotone functions. In Section 4, we prove the case where $N = d_x = d_y \geq 2$. The construction is based on the approximation power of neural ODEs. In Section 5, we consider the case where $d_x \neq d_y$ and discuss the case of more general activation functions. Finally, we conclude the paper in Section 6. All formal proofs of the results are presented in the Appendix.

### 2 MAIN RESULTS

In this paper, we consider the standard feed-forward neural network with $N$ neurons at each hidden layer. We say that a $\sigma$ network with depth $L$ is a function with inputs $x \in \mathbb{R}^{d_x}$ and outputs $y \in \mathbb{R}^{d_y}$, which has the following form:

$$y \equiv f_L(x) = W_{L+1}\sigma(W_L(\cdots\sigma(W_1 x + b_1) + \cdots) + b_L) + b_{L+1}, \tag{1}$$

where $b_i$ are bias vectors, $W_i$ are weight matrices, and $\sigma(\cdot)$ is the activation function. For the case of multiple activation functions, for instance, $\sigma_1$ and $\sigma_2$, we call $f_L$ a $\sigma_1+\sigma_2$ network. In this situation, the activation function of each neuron is either $\sigma_1$ or $\sigma_2$. In this paper, we consider arbitrary activation functions, while the following activation functions are emphasized: ReLU $(\max(x,0))$, leaky-ReLU $(\max(x, \alpha x), \alpha \in (0, 1)$ is a fixed positive

parameter), ABS ($|x|$), SIN ($\sin(x)$), STEP ($1_{x>0}$), FLOOR ($\lfloor x \rfloor$) and UOE (*universal ordering of extrema*, which will be defined later).

**Lemma 1.** *For any compact domain $\mathcal{K} \subset \mathbb{R}^{d_x}$ and any finite set of activation functions $\{\sigma_i\}$, the $\{\sigma_i\}$ networks with width $w < w_{\min}^* \equiv \max(d_x, d_y)$ do not have the UAP for both $L^p(\mathcal{K}, \mathbb{R}^{d_y})$ and $C(\mathcal{K}, \mathbb{R}^{d_y})$.*

**$L^p$-UAP and $C$-UAP.** The lemma indicates that $w_{\min}^* \equiv \max(d_x, d_y)$ is a universal lower bound for the UAP in both $L^p(\mathcal{K}, \mathbb{R}^{d_y})$ and $C(\mathcal{K}, \mathbb{R}^{d_y})$. The main result of this paper illustrates that the minimal width $w_{\min}^*$ can be achieved. We consider the UAP for these two function classes, *i.e.*, $L^p$-UAP and $C$-UAP, respectively. Note that any compact domain can be covered by a big cubic, the functions on the former can be extended to the latter, and the cubic can be mapped to the unit cubic by a linear function. This allows us to assume $\mathcal{K}$ to be a (unit) cubic without loss of generality.

## 2.1 $L^p$-UAP

**Theorem 2.** *Let $\mathcal{K} \subset \mathbb{R}^{d_x}$ be a compact set; then, for the function class $L^p(\mathcal{K}, \mathbb{R}^{d_y})$, the minimum width of leaky-ReLU networks having $L^p$-UAP is exactly $w_{\min} = \max(d_x, d_y, 2)$.*

The theorem indicates that leaky-ReLU networks achieve the critical width $w_{\min}^* = \max(d_x, d_y)$, except for the case of $d_x = d_y = 1$. The idea is to consider the case where $d_x = d_y = d > 1$ and let the network width equal $d$. According to the results of Duan et al. (2022), leaky-ReLU networks can approximate the flow map of neural ODEs. Thus, we can use the approximation power of neural ODEs to finish the proof. Li et al. (2022) proved that many neural ODEs could approximate continuous functions in the $L^p$ norm. This is based on the fact that orientation preserving diffeomorphisms can approximate continuous functions Brenier & Gangbo (2003).

The exclusion of dimension one is because of the monotonicity of leaky ReLU. When we add a nonmonotone activation function such as the absolute value function or sine function, the $L^p$-UAP at dimension one can be achieved.

**Theorem 3.** *Let $\mathcal{K} \subset \mathbb{R}^{d_x}$ be a compact set; then, for the function class $L^p(\mathcal{K}, \mathbb{R}^{d_y})$, the minimum width of leaky-ReLU+ABS networks having $L^p$-UAP is exactly $w_{\min} = \max(d_x, d_y)$.*

## 2.2 $C$-UAP

$C$-UAP is more demanding than $L^p$-UAP. However, if the activation functions could include discontinuous functions, the same critical width $w_{\min}^*$ can be achieved. Following the encoder-memory-decoder approach in Park et al. (2021), the step function is replaced by the floor function, and one can obtain the minimal width $w_{\min} = \max(d_x, 2, d_y)$.

**Lemma 4.** *Let $\mathcal{K} \subset \mathbb{R}^{d_x}$ be a compact set; then, for the function class $C(\mathcal{K}, \mathbb{R}^{d_y})$, the minimum width of ReLU+FLOOR networks having $C$-UAP is exactly $w_{\min} = \max(d_x, 2, d_y)$.*

Since ReLU and FLOOR are monotone functions, the $C$-UAP critical width $w_{\min}^*$ does not hold for $C([0, 1], \mathbb{R})$. This seems to be the case even if we add ABS or SIN as an additional activator. However, it is still possible to use the UOE function (Definition 12).

**Theorem 5.** *The UOE networks with width $d_y$ have $C$-UAP for functions in $C([0, 1], \mathbb{R}^{d_y})$.*

**Corollary 6.** *Let $\mathcal{K} \subset \mathbb{R}^{d_x}$ be a compact set; then, for the continuous function class $C(\mathcal{K}, \mathbb{R}^{d_y})$, the minimum width of UOE+FLOOR networks having $C$-UAP is exactly $w_{\min} = \max(d_x, d_y)$.*

## 3 APPROXIMATION IN DIMENSION ONE ($N = d_x = d_y = d = 1$)

In this section, we consider one-dimensional functions and neural networks with a width of one. In this case, the expression of ReLU networks is extremely poor. Therefore, we consider the leaky ReLU activation $\sigma_\alpha(x)$ with a fixed parameter $\alpha \in (0, 1)$. Note that leaky-ReLU is strictly monotonic, and it was proven by Duan et al. (2022) that any monotone function in

$C([0,1],\mathbb{R})$ can be uniformly approximated by leaky-ReLU networks with width one. This is useful for our construction to approximate nonmonotone functions. Since the composition of monotone functions is also a monotone function, to approximate nonmonotone functions we need to add a nonmonotone activation function.

Let us consider simple nonmonotone functions, such as $|x|$ or $\sin(x)$. We show that leaky-ReLU+ABS or leaky-ReLU+SIN can approximate any continuous function $f^*(x)$ under the $L^p$ norm. The idea, shown in Figure 1, is that the target function $f^*(x)$ can be uniformly approximated by the polynomial $p(x)$, which can be represented as the composition

$$g \circ u(x) = p(x) \approx f^*(x).$$

Here, the outer function $g(x)$ is any continuous function whose value at extrema matches the value at extrema of $p(x)$, and the inner function $u(x)$ is monotonically increasing, which adjusts the location of the extrema (see Figure 1). Since polynomials have a finite number of extrema, the inner function $u(x)$ is piecewise continuous.

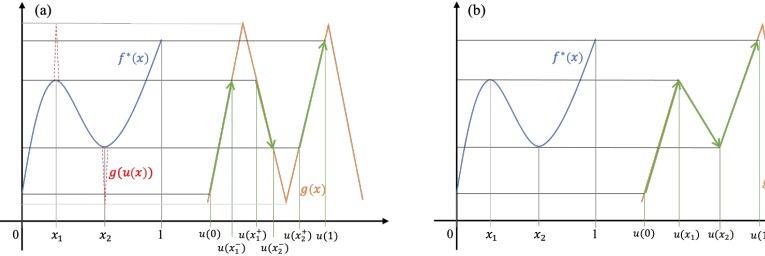

Figure 1: Example of approximating/representing a polynomial by the composition of a monotonically increasing function $u(x)$ and a nonmonotone function $g(x)$. (a) only matching the ordering of extrema values, (b) matching the values as well.

For $L^p$-UAP, the approximation is allowed to have a large deviation on a small interval; therefore, the extrema could not be matched exactly (over a small error). For example, we can choose $g(x)$ as the sine function or the sawtooth function (which can be approximated by ABS networks), and $u(x)$ is a leaky-ReLU network approximating $g^{-1} \circ p(x)$ at each monotone interval of $p$. Figure 1(a) shows an example of the composition.

For $C$-UAP, matching the extrema while keeping the error small is needed. To achieve this aim, we introduce the UOE functions.

**Definition 7** (Universal ordering of extrema (UOE) functions). *A UOE function is a continuous function in $C(\mathbb{R}, \mathbb{R})$ such that any (finite number of) possible ordering(s) of values at the (finite) extrema can be found in the extrema of the function.*

There are an infinite number of UOE functions. Here, we give an example, as shown in Figure 2. This UOE function $\rho(x)$ is defined by a sequence $\{o_i\}_{i=1}^{\infty}$,

$$\rho(x) = \begin{cases} x/4, & x \leq 0, \\ o_i + (x-i)(o_{i+1} - o_i), & x \in [i, i+1), \end{cases} \tag{2}$$

where $\{o_i\}_{i=1}^{\infty} = (\underline{1,2},\underline{2,1},\underline{1,2,3},\underline{1,3,2},\underline{2,1,3},\underline{2,3,1},\underline{3,1,2},\underline{3,2,1},\underline{1,2,3,4},...)$ is the concatenation of all permutations of positive integer numbers. The term UOE in this paper means this function $\rho$. Since the UOE function $\rho(x)$ can represent leaky-ReLU $\sigma_{1/4}$ on any finite interval, this implies that the UOE networks can uniformly approximate any monotone functions.

To illustrate the $C$-UAP of UOE networks, we only need to construct a continuous function $g(x)$ matching the extrema of $p(x)$ (see Figure 1(b)). That is, construct $g(x)$ by the composition $\tilde{u} \circ \rho(x)$, where $\tilde{u}(x)$ is a monotone and continuous function. This is possible since the UOE function contains any ordering of the extrema.

The following lemma summarizes the approximation of one-dimensional functions. As a consequence, Theorem 5 holds since functions in $C([0,1], \mathbb{R}^{d_y})$ can be regarded as $d_y$ one-dimensional functions.

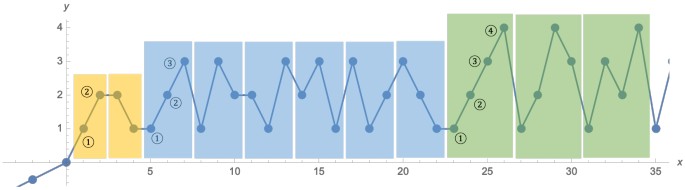

Figure 2: An example of the UOE function $\rho(x)$, which has an infinite number of pieces.

**Lemma 8.** *For any function $f^*(x) \in C[0,1]$ and $\varepsilon > 0$, there is a leaky-ReLU+ABS (or leaky-ReLU+SIN) network with width one and depth $L$ such that $\int_0^1 |f^*(x) - f_L(x)|^p dx < \varepsilon^p$. There is a leaky-ReLU+UOE network with a width of one and a depth of $L$ such that $|f^*(x) - f_L(x)| < \varepsilon, \forall x \in [0,1]$.*

## 4  CONNECTION TO THE NEURAL ODES ($N = d_x = d_y = d \geq 2$)

Now, we turn to the high-dimensional case and connect the feed-forward neural networks to neural ODEs. To build this connection, we assume that the input and output have the same dimension, $d_x = d_y = d$.

Consider the following neural ODE with one-hidden layer neural fields:

$$\begin{cases} \dot{x}(t) = v(x(t), t) := A(t) \tanh(W(t)x(t) + b(t)), t \in (0, \tau), \\ x(0) = x_0, \end{cases} \quad (3)$$

where $x, x_0, \in \mathbb{R}^d$ and the time-dependent parameters $(A, W, b) \in \mathbb{R}^{d \times d} \times \mathbb{R}^{d \times d} \times \mathbb{R}^d$ are piecewise constant functions of $t$. The flow map is denoted as $\phi^\tau(\cdot)$, which is the function from $x_0$ to $x(\tau)$. According to the approximation results of neural ODEs (see Li et al. (2022); Tabuada & Gharesifard (2020); Ruiz-Balet & Zuazua (2021) for examples), we have the following lemma.

**Lemma 9** (Special case of Li et al. (2022) )**.** *Let $d \geq 2$. Then, for any continuous function $f^* : \mathbb{R}^d \to \mathbb{R}^d$, any compact set $\mathcal{K} \subset \mathbb{R}^d$, and any $\varepsilon > 0$, there exist a time $\tau \in \mathbb{R}^+$ and a piecewise constant input $(A, W, b) : [0, \tau] \to \mathbb{R}^{d \times d} \times \mathbb{R}^{d \times d} \times \mathbb{R}^d$ so that the flow-map $\phi^\tau$ associated with the neural ODE (3) satisfies: $||f^* - \phi^\tau||_{L^p(\mathcal{K})} \leq \varepsilon$.*

Next, we consider the approximation of the flow map associated with (3) by neural networks. Recently, Duan et al. (2022) found that leaky-ReLU networks could perform such approximations.

**Lemma 10** (Theorem 2.2 in Duan et al. (2022))**.** *If the parameters $(A, W, b)$ in (3) are piecewise constants, then for any compact set $\mathcal{K}$ and any $\varepsilon > 0$, there is a leaky-ReLU network $f_L(x)$ with width $d$ and depth $L$ such that*

$$\|\phi^\tau(x) - f_L(x)\| \leq \varepsilon, \forall x \in \mathcal{K}. \quad (4)$$

Combining these two lemmas, one can directly prove the following corollary, which is a part of our Theorem 2.

**Corollary 11.** *Let $\mathcal{K} \subset \mathbb{R}^d$ be a compact set and $d \geq 2$; then, for the function class $L^p(\mathcal{K}, \mathbb{R}^d)$, the leaky-ReLU networks with width $d$ have $L^p$-UAP.*

Here, we summarize the main ideas of this result. Let us start with the discretization of the ODE by the splitting approach (see McLachlan & Quispel (2002) for example). Consider the spliting of (3) with $v(x, t) = \sum_{i,j} v_i^{(j)}(x, t) e_j$, where $v_i^{(j)}(x, t) = A_{ji}(t) \tanh(W_{i,:}(t)x + b_i(t))$ is a scalar function and $e_j$ is the $j$-th axis unit vector. Then for a given time step $\Delta t = \tau/K$, ($K$ large enough), the splitting method gives the following iteration of $x_k$ which approximates $\phi^{k\Delta t}(x_0)$,

$$x_{k+1} = T_k^{(d,d)} \circ \cdots \circ T_k^{(1,2)} \circ T_k^{(1,1)} x_k, \quad (5)$$

where the map $T_k^{(i,j)} : x \to y$ is defined as

$$
\begin{cases}
y^{(l)} = x^{(l)}, l \neq j, \\
y^{(j)} = x^{(j)} + \Delta t v_i^{(j)}(x, k\Delta t) = x^{(j)} + a\Delta t \tanh(wx + \beta).
\end{cases} \tag{6}
$$

Here the superscript in $x^{(l)}$ means the $l$-th coordinate of $x$. $a = A_{ji}, w = W_{i,:}$ and $\beta = b_i$ take their value at $t = k\Delta t$. Note that the scalar functions $\tanh(\xi)$ and $\xi + a\Delta t \tanh(\xi)$ are monotone with respect to $\xi$ when $\Delta t$ is small enough. This allows us to construct leaky-ReLU networks with width $d$ to approximate each map $T_k^{(i,j)}$ and then approximate the flow-map, $\phi^\tau(x_0) \approx x_K$.

Note that Lemma 10 holds for all dimensions, while Lemma 9 holds for dimensions larger than one. This is because flow maps are orientation-preserving diffeomorphisms, and they can approximate continuous functions only for dimensions larger than one; see Brenier & Gangbo (2003). The approximation is based on control theory where the flow map can be adjusted to match any finite set of input-output pairs. This match does not hold for dimension one. However, the case of dimension one is discussed in the last section.

## 5    ACHIEVING THE MINIMAL WIDTH

Now, we turn to the cases where the input and output dimensions cannot be equal.

### 5.1    UNIVERSAL LOWER BOUND $w_{\min}^* = \max(d_x, d_y)$

Here, we give a sketch of the proof of Lemma 1, which states that $w_{\min}^*$ is a universal lower bound over all activation functions. Parts of Lemma 1 have been demonstrated in many papers, such as Park et al. (2021). Here, we give proof by two counterexamples that are simple and easy to understand from the topological perspective. It contains two cases: 1) there is a function $f^*$ that cannot be approximated by networks with width $w \leq d_x - 1$; 2) there is a function $f^*$ that cannot be approximated by networks with width $w \leq d_y - 1$. Figure 3(a)-(b) shows the counterexamples that illustrate the essence of the proof.

For the first case, $w \leq d_x - 1$, we show that $f^*(x) = \|x\|^2, x \in \mathcal{K} = [-2, 2]^{d_x}$, is what we want; see Figure 3(a). In fact, we can relax the networks to a function $f(x) = \phi(Wx + b)$, where $Wx + b$ is a transformer from $\mathbb{R}^{d_x}$ to $\mathbb{R}^{d_x - 1}$ and $\phi(x)$ could be any function. A consequence is that there exists a direction $v$ (set as the vector satisfying $Wv = 0, \|v\| = 1$) such that $f(x) = f(x + \lambda v)$ for all $\lambda \in \mathbb{R}$. Then, considering the sets $A = \{x : \|x\| \leq 0.1\}$ and $B = \{x : \|x - v\| \leq 0.1\}$, we have

$$
\begin{aligned}
\int_{\mathcal{K}} |f(x) - f^*(x)| dx &\geq \int_A |f(x) - f^*(x)| dx + \int_B |f(x) - f^*(x)| dx \\
&\geq \int_A (|f(x) - f^*(x)| + |f(x+v) - f^*(x+v)|) dx \\
&\geq \int_A (|f^*(x) - f^*(x+v)|) dx \geq 0.8|A|.
\end{aligned}
$$

Since the volume of $A$ is a fixed positive number, the inequality implies that even the $L^1$ approximation for $f^*$ is impossible. The case of the $L^p$ norm and the uniform norm is impossible as well.

For the second case, $w \leq d_y - 1$, we show the example of $f^*$, which is the parametrized curve from $\mathbf{0}$ to $\mathbf{1}$ along the edge of the cubic, see Figure 3(b). Relaxing the networks to a function $f(x) = W\psi(x) + b$, $\psi(x)$ could be any function. Since the range of $f$ is in a hyperplane while $f^*$ has a positive distance to any hyperplane, the target $f^*$ cannot be approximated.

### 5.2    ACHIEVING $w_{\min}^*$ FOR $L^p$-UAP

Now, we show that the lower bound $w_{\min}^*$ for $L^p$-UAP can be achieved by leaky-ReLU+ABS networks. Without loss of generality, we consider $\mathcal{K} = [0, 1]^{d_x}$.

For any function $f^*$ in $L^p([0,1]^{d_x}, \mathbb{R}^{d_y})$, we can extend it to a function $\tilde{f}^*$ in $L^p([0,1]^d, \mathbb{R}^d)$ by filling in zeros where $d = \max(d_x, d_y) = w^*_{\min}$. When $d_x > 1$ or $d_y > 1$, the $L^p$-UAP for leaky-ReLU networks with width $w^*_{\min}$ is obtained by using Corollary 11. Recall that by the Lemma 1, $w^*_{\min}$ is optimal, and we obtain our main result Theorem 2.

Combining the case of $d_x = d_y = d = 1$ in Section 3, adding absolute function ABS as an additional activation function, we obtain Theorem 3.

## 5.3 Achieving $w^*_{\min}$ for $C$-UAP

Here, we use the encoder-memorizer-decoder approach proposed in Park et al. (2021) to achieve the minimum width. Without loss of generality, we consider the function class $C([0,1]^{d_x}, [0,1]^{d_y})$. The encoder-memorizer-decoder approach includes three parts:

1) an **encoder** maps $[0,1]^{d_x}$ to $[0,1]$ which quantizes each coordinate of $x$ by a $K$-bit binary representation and concatenates the quantized coordinates into a single scalar value $\bar{x}$ having a $(d_x K)$-bit binary representation;

2) a **memorizer** maps each codeword $\bar{x}$ to its target codeword $\bar{y}$;

3) a **decoder** maps $\bar{y}$ to the quantized target that approximates the true target.

As illustrated in Figure 3(c), using the floor function instead of a step function, one can construct the encoder by FLOOR networks with width $d_x$ and the decoder by FLOOR networks with width $d_y$. The memorizer is a one-dimensional scalar function that can be approximated by ReLU networks with a width of two or UOE networks with a width of one. Therefore, the minimal widths $\max(d_x, 2, d_y)$ and $\max(d_x, d_y)$ are obtained, which demonstrate Lemma 4 and Corollary 6, respectively.

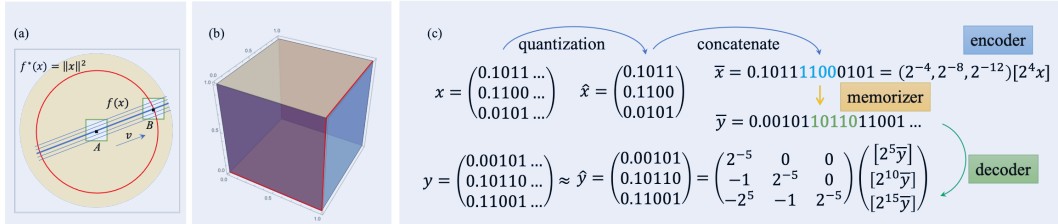

Figure 3: (a)(b) Counterexamples for proving Lemma 1. (a) Points $A$ and $B$ on a level set of networks $f(x)$; $f(A) = f(B)$ but $f^*(A) - f^*(B)$ is not small. (b) The curve from $\mathbf{0}$ to $\mathbf{1}$ along the edge of the cubic has a positive distance to any hyperplane. (c) illustration of the encoder-memorizer-decoder scheme for $C$-UAP by an example where $d_x = d_y = 3$, 4 bits for the input and 5 bits for the output.

## 5.4 Effect of the activation functions

Here, we emphasize that our universal bound of the minimal width is optimized over arbitrary activation functions. However, it cannot always be achieved when the activation functions are fixed. Here, we discuss the case of monotone activation functions.

If the activation functions are strictly monotone and continuous (such as leaky-ReLU), a width of at least $d_x + 1$ is needed for $C$-UAP. This can be understood through topology theory. Leaky-ReLU, the nonsingular linear transformer, and its inverse are continuous and homeomorphic. Since compositions of homeomorphisms are also homeomorphisms, we have the following proposition: If $N = d_x = d_y = d$ and the weight matrix in leaky-ReLU networks are nonsingular, then the input-output map is a homeomorphism. Note that singular matrices can be approximated by nonsingular matrices; therefore, we can restrict the weight matrix in neural networks to the nonsingular case.

When $d_x \geq d_y$, we can reformulate the leaky-ReLU network as $f_L(x) = W_{L+1}\psi(x) + b_{L+1}$, where $\psi(x)$ is the homeomorphism. Note that considering the case where $d_y = 1$ is sufficient,

according to Hanin & Sellke (2017); Johnson (2019). They proved that the neural network width $d_x$ cannot approximate any scalar function with a level set containing a bounded path component. This can be easily understood from the perspective of topology theory. An example is to consider the function $f^*(x) = \|x\|^2, x \in \mathcal{K} = [-2, 2]^{d_x}$ shown in Figure 4.

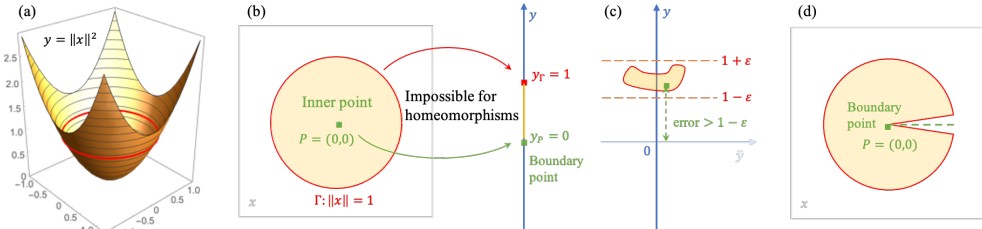

Figure 4: Illustrating the possibility of UAP when $N = d_x$. (a) Plot of $f^*(x) = \|x\|^2$ and its contour at $\|x\| = 1$. (b) The original point $P$ is an inner point of the unit ball, while its image is a boundary point, which is impossible for homeomorphisms. (c) Any homeomorphism, approximating $\|x\|^2$ with error less than $\varepsilon$ (=0.1 for example) on $\Gamma$, should have error larger than $1 - \varepsilon$ (=0.9) at $P$. (d) Approximating $f^*$ in $L^p$ is possible by leaving a small region.

The case where $d_x < d_y$. We present a simple example in Figure 5. The curve '4' corresponding to a continuous function from $[0, 1] \subset \mathbb{R}$ to $\mathbb{R}^2$ cannot be uniformly approximated. However, the $L^p$ approximation is still possible.

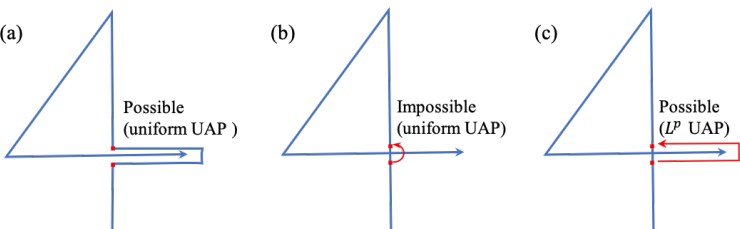

Figure 5: Illustrating the possibility of $C$-UAP when $d_x \leq d_y$. The curve in (a) is homeomorphic to the interval $[0, 1]$, while the curve '4' in (b) is not and cannot be approximated uniformly by homeomorphisms. The $L^p$ approximation is possible via (a).

## 6    CONCLUSION

Let us summarize the main results and implications of this paper. After giving the universal lower bound of the minimum width for the UAP, we proved that the bound is optimal by constructing neural networks with some activation functions.

For the $L^p$-UAP, our construction to achieve the critical width was based on the approximation power of neural ODEs, which bridges the feed-forward networks to the flow maps corresponding to the ODEs. This allowed us to understand the UAP of the FNN through topology theory. Moreover, we obtained not only the lower bound but also the upper bound.

For the $C$-UAP, our construction was based on the encoder-memorizer-decoder approach in Park et al. (2021), where the activation sets contain a discontinuous function $\lfloor x \rfloor$. It is still an open question whether we can achieve the critical width by continuous activation functions. Johnson (2019) proved that continuous and monotone activation functions need at least width $d_x + 1$. This implies that nonmonotone activation functions are needed. By using the UOE activation, we calculated the critical width for the case of $d_x = 1$. It would be of interest to study the case of $d_x \geq 2$ in future research.

We remark that our UAP is for functions on a compact domain. Examining the critical width of the UAP for functions on unbounded domains is desirable for future research.

ACKNOWLEDGMENTS

We thank anonymous reviewers for their valuable comments and useful suggestions. This research is supported by the National Natural Science Foundation of China (Grant No. 12201053).

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

# A  PROOF OF THE LEMMAS

## A.1  PROOF OF LEMMA 8

We give a definition and a lemma below that are useful for proving Lemma 8.

**Definition 12.** *We say two functions, $f_1, f_2 \in C(\mathbb{R}, \mathbb{R})$, have the same ordering of extrema if they have the following properties:*

*1) $f_i(x)$ has only a finite number of extrema that are (increasing) $x_{i,j}^*, j = 1, 2, ..., m_i$.*

*2) $m_1 = m_2 =: m$ and the two sequences,*

$$S_1 := \{f_1(-\infty), f_1(x_{1,1}^*), ..., f_1(x_{1,m}^*), f_1(+\infty)\},$$

*and*

$$S_2 := \{f_2(-\infty), f_2(x_{2,1}^*), ..., f_2(x_{2,m}^*), f_2(+\infty)\},$$

*have the same ordering,* i.e.,

$$S_{1,i} < S_{1,j} \iff S_{2,i} < S_{2,j}, \quad \forall i, j,$$
$$S_{1,i} = S_{1,j} \iff S_{2,i} = S_{2,j}, \quad \forall i, j.$$

**Lemma 13.** *Let $f_1$ and $f_2$ be continuous functions in $C(\mathbb{R}, \mathbb{R})$ that have the same ordering of extrema; then, there are two strictly monotone functions, $v$ and $u$, such that*

$$f_1 = v \circ f_2 \circ u.$$

*Proof.* Here, we use the same notation in Definition 12. The functions $v$ and $u$ can be constructed as follows.

(1) Construct the outer function $v$ that tries to match the function values at the extrema. The only requirement is that
$$S_{1,i} = v(S_{2,i}), \quad \forall i.$$
Since $S_1$ and $S_2$ have the same ordering, it is easy to construct such a function $v$ that is continuous and strictly increasing, for example, piecewise linear.

(2) Construct the inner function $u$ to match the location of the extrema. Denote $g = v \circ f_2$, which satisfies $f_1(x_{1,i}^*) = g(x_{2,i}^*)$. Since $f_1$ and $g$ are strictly monotone and continuous on the intervals $I_i := (x_{1,i}^*, x_{1,i+1}^*)$ and $J_i = (x_{2,i}^*, x_{2,i+1}^*)$, respectively, we can construct the function $u$ on $I_i$ as

$$u(x) = g^{-1}(f_1(x)), x \in I_i.$$

Combining each piece of $u$, we have a strictly increasing and continuous function $u$ on the whole space $\mathbb{R}$. As a consequence, we have $f_1 = g \circ u = v \circ f_2 \circ u$.

$\square$

**Lemma 8.** *For any function $f^*(x) \in C[0, 1]$ and $\varepsilon > 0$,*

*1) there is a leaky-ReLU+ABS (or leaky-ReLU+SIN) network with width one and depth $L$ such that $\int_0^1 |f^*(x) - f_L(x)|^p dx < \varepsilon^p$.*

*2) there is a leaky-ReLU + UOE network width one and depth $L$ such that $|f^*(x) - f_L(x)| < \varepsilon, \forall x \in [0, 1]$.*

*Proof.* We mainly provide proof of the second point, while the first point can be proven using the same scheme.

For any function $f^*(x) \in C([0, 1], \mathbb{R})$ and $\varepsilon > 0$, we can approximate it by a polynomial $p_n(x)$ with order $n$ such that

$$|f^*(x) - p_n(x)| \leq \varepsilon/2, \quad \forall x \in [0, 1],$$

according to the well-known Weierstrass approximation theorem. Without a loss of generality, we can assume that $p_n(x)$ is not the same at all of its extrema. Then, we can represent $p_n(x)$ by the following composition, using Lemma 13 and the property of UOE:

$$p_n(x) = v \circ \rho \circ u(x), \tag{7}$$

where $\rho(x)$ is the UOE function (2) and $v(x)$ and $u(x)$ are monotonically increasing continuous functions.

Then, we can approximate $p_n(x)$ by UOE networks. Since $v(x)$ and $u(x)$ are monotone, there are UOE networks $\tilde{v}(x)$ and $\tilde{u}(x)$ such that $\|v - \tilde{v}\|$ and $\|u - \tilde{u}\|$ are arbitrarily small. Hence, there is a UOE network $f_L(x) = \tilde{v} \circ \rho \circ \tilde{u}(x)$ that can approximate $p_n(x)$ such that

$$|p_n(x) - f_L(x)| \le \varepsilon/2, \quad \forall x \in [0, 1],$$

which implies that

$$|f^*(x) - f_L(x)| \le \varepsilon.$$

This completes the proof of the second point.

For the first point, we only emphasize that it is easy to construct a function $f(x)$ that has the same local maximum and local minimum in the interval and has $\|f - f^*\|_{L^p}$ small enough. This $f(x)$ has the same ordering of extrema as the sawtooth function (or sine) and hence can be uniformly approximated by leaky-ReLU+ABS (or leaky-ReLU+SIN) networks $f_L$. As a consequence, $\|f_L - f^*\|_{L^p}$ is small enough. □

## A.2 Proof of Lemma 9

**Lemma 9.** *Let $d \ge 2$. Then, for any continuous function $f^* : \mathbb{R}^d \to \mathbb{R}^d$, any compact set $\mathcal{K} \subset \mathbb{R}^d$, and any $\varepsilon > 0$, there exist a time $\tau \in \mathbb{R}^+$ and a piecewise constant input $(A, W, b) : [0, \tau] \to \mathbb{R}^{d \times d} \times \mathbb{R}^{d \times d} \times \mathbb{R}^d$ so that the flow map $\phi^\tau$ associated with the neural ODE (3) satisfies: $\|f^* - \phi^\tau\|_{L^p(\mathcal{K})} \le \varepsilon$.*

*Proof.* This is a special case of Theorem 2.3 in Li et al. (2022). □

## A.3 Proof of Lemma 10

**Lemma 10.** *If the parameters $(A, W, b)$ in (3) are piecewise constants, then for any compact set $\mathcal{K}$ and any $\varepsilon > 0$, there is a leaky-ReLU network $f_L(x)$ with width $d$ and depth $L$ such that*

$$\|\phi^\tau(x) - f_L(x)\| \le \varepsilon, \forall x \in \mathcal{K}. \tag{8}$$

*Proof.* It is Theorem 2.2 in Duan et al. (2022). □

## A.4 Proof of Corollary 11

**Corollary 11.** *Let $\mathcal{K} \subset \mathbb{R}^d$ be a compact set and $d \ge 2$; then, for the function class $L^p(\mathcal{K}, \mathbb{R}^d)$, the leaky-ReLU networks with width $d$ have $L^p$-UAP.*

*Proof.* For any $f^*(x) \in L^p(\mathcal{K}, \mathbb{R}^d)$ and $\varepsilon > 0$, there is a flow map $\phi^\tau(x)$ associated with the neural ODE (3) such that (according to Lemma 9)

$$\|f^*(\cdot) - \phi^\tau(\cdot)\|_{L^p} \le \frac{\varepsilon}{2}.$$

Then, employing Lemma 10, there is a leaky-ReLU network $f_L$ such that

$$\|f_L(\cdot) - \phi^\tau(\cdot)\|_{L^p} \le \frac{\varepsilon}{2}.$$

Therefore, we have

$$\|f_L(\cdot) - f^*(\cdot)\|_{L^p} \le \|f^*(\cdot) - \phi^\tau(\cdot)\|_{L^p} + \|f_L(\cdot) - \phi^\tau(\cdot)\|_{L^p} \le \varepsilon.$$

□

## B    Proof of the main results

### B.1    Proof of Lemma 1

**Lemma 1.** *For any compact domain $\mathcal{K} \subset \mathbb{R}^{d_x}$ and any finite set of activation functions $\{\sigma_i\}$, the $\{\sigma_i\}$ networks with width $w < w_{\min}^* \equiv \max(d_x, d_y)$ do not have the UAP for both $L^p(\mathcal{K}, \mathbb{R}^{d_y})$ and $C(\mathcal{K}, \mathbb{R}^{d_y})$.*

*Proof.* It is enough to show the following two counterexamples $f^*(x)$ that cannot be approximated in the $L^p$-norm.

1) $f^*(x) = \|x\|^2, x \in \mathcal{K} = [-2, 2]^{d_x}$, cannot be approximated by any networks with widths less than $d_x - 1$. In fact, we can relax the networks to a function $f(x) = \phi(Wx + b)$, where $Wx + b$ is a transformer from $\mathbb{R}^{d_x}$ to $\mathbb{R}^{d_x - 1}$ and $\phi(x)$ could be any function. A consequence is that there exists a direction $v$ (set as the vector satisfying $Wv = 0$, $\|v\| = 1$) such that $f(x) = f(x + \lambda v)$ for all $\lambda \in \mathbb{R}$. Then, considering the sets $A = \{x : \|x\| \le 0.1\}$ and $B = \{x : \|x - v\| \le 0.1\}$, we have

$$
\begin{aligned}
\int_{\mathcal{K}} |f(x) - f^*(x)| dx &\ge \int_A |f(x) - f^*(x)| dx + \int_B |f(x) - f^*(x)| dx \\
&\ge \int_A (|f(x) - f^*(x)| + |f(x + v) - f^*(x + v)|) dx \\
&\ge \int_A (|f^*(x) - f^*(x + v)|) dx \ge 0.8 |A|.
\end{aligned}
$$

Since the volume of $A$ is a fixed positive number, the inequality implies that even the $L^1$ approximation for $f^*$ is impossible. The case of the $L^p$ norm and the uniform norm is impossible as well.

2) The function $f^*$, the parametrized curve from $\mathbf{0}$ to $\mathbf{1}$ along the edge of the cubic, cannot be approximated by any networks with a width less than $d_y - 1$. Relaxing the networks to a function $f(x) = W\psi(x) + b$, $\psi(x)$ could be any function. Since the range of $f$ is in a hyperplane while $f^*$ has a positive distance to any hyperplane, the target $f^*$ cannot be approximated.    $\square$

### B.2    Proof of Theorem 2

**Theorem 2.** *Let $\mathcal{K} \subset \mathbb{R}^{d_x}$ be a compact set; then, for the function class $L^p(\mathcal{K}, \mathbb{R}^{d_y})$, the minimum width of leaky-ReLU networks having $L^p$-UAP is exactly $w_{\min} = \max(d_x, d_y, 2)$.*

*Proof.* Using Lemma 1, we only need to prove two points: 1) the $L^p$-UAP holds when $\max(d_x, d_y) \ge 2$, 2) when $d_x = d_y = 1$, there is a function that cannot be approximated by leaky-ReLU networks with width one (since width two is enough for the $L^p$-UAP).

The first point is a consequence of Corollary 11 since we can extend the target function to dimension $d = \max(d_x, d_y)$.

The second point is obvious since leaky-ReLU networks with a width of one are monotone functions that cannot approximate nonmonotone functions such as $f^*(x) = x^2, x \in [-1, 1]$.    $\square$

### B.3    Proof of Theorem 3

**Theorem 3.** *Let $\mathcal{K} \subset \mathbb{R}^{d_x}$ be a compact set; then, for the function class $L^p(\mathcal{K}, \mathbb{R}^{d_y})$, the minimum width of leaky-ReLU+ABS networks having $L^p$-UAP is exactly $w_{\min} = \max(d_x, d_y)$.*

*Proof.* This is a consequence of Theorem 2 (for the case of $\max(d_x, d_y) \ge 2$) combined with Lemma 8 (for the case of $d_x = d_y = 1$).    $\square$

### B.4 PROOF OF LEMMA 4

**Lemma 4.** *Let $\mathcal{K} \subset \mathbb{R}^{d_x}$ be a compact set; then, for the function class $C(\mathcal{K}, \mathbb{R}^{d_y})$, the minimum width of ReLU+FLOOR networks having C-UAP is exactly $w_{\min} = \max(d_x, 2, d_y)$.*

*Proof.* Recalling the results of Lemma 1, we only need to prove two points: 1) the C-UAP holds when $\max(d_x, d_y) \geq 2$, 2) when $d_x = d_y = 1$, there is a function that cannot be approximated by ReLU+FLOOR networks with width one (since width two is enough for the $C$-UAP).

The first step can be constructed by the encoder-memorizer-decoder approach. The second point is obvious since ReLU+FLOOR networks with width one are monotone functions that cannot approximate nonmonotone functions such as $f^*(x) = x^2, x \in [-1, 1]$. □

### B.5 PROOF OF THEOREM 5

**Theorem 5.** *The UOE networks with width $d_y$ have C-UAP for functions in $C([0,1], \mathbb{R}^{d_y})$.*

*Proof.* Since functions in $C([0,1], \mathbb{R}^{d_y})$ can be regarded as $d_y$ one-dimensional functions, it is enough to prove the case of $d_y = 1$, which is the result in Lemma 8. □

### B.6 PROOF OF COROLLARY 6

**Corollary 6.** *Let $\mathcal{K} \subset \mathbb{R}^{d_x}$ be a compact set; then, for the continuous function class $C(\mathcal{K}, \mathbb{R}^{d_y})$, the minimum width of UOE+FLOOR networks having C-UAP is exactly $w_{\min} = \max(d_x, d_y)$.*

*Proof.* The case where $\max(d_x, d_y) \geq 2$ is a consequence of Lemma 4 since the UOE function contains the leaky-ReLU as a part. The case where $\max(d_x, d_y) = 1$, *i.e.* $d_x = d_y = 1$, is a consequence of Lemma 8. □

