# OpenReview forum: "Achieve the Minimum Width of Neural Networks for Universal Approximation"
_ICLR.cc/2023/Conference — ICLR 2023 poster_

### Official Review · Reviewer_emi4 · 2022-10-24

**Confidence:** 2
**Correctness:** 3
**Technical Novelty And Significance:** 3
**Empirical Novelty And Significance:** 3
**Recommendation:** 6

**Clarity, Quality, Novelty And Reproducibility:**

Clarity: This paper is clearly written and well organized. I find it easy to follow.
Quality: This paper is technically sound.
Novelty: The novelty of this paper is high.


**Strength And Weaknesses:**

The related works are adequately cited. The main results in this paper will certainly help us have a better understating of the universal approximation property of deep neural networks from a theoretical way. I have checked the technique parts and found that the proofs are solid. The main result, which states that for any activation functions, the minimum width of a DNN to approximate any functions in $L^p(\mathcal K, \mathbb{R}^{d_y})$ and $C(\mathcal K, \mathbb{R}^{d_y})$ is at least $\max (d_x, d_y)$, is an unexpected and elegant result, which is a non-trivial extension of previous results in this field. It would be more interesting if the authors could study the exact minimum width for more activation functions and more architectures used in practice.

**Summary Of The Paper:**

The authors study the universal approximation problem of functions using neural networks. They showed that, the minimum width to approximate any functions in $L^p(\mathcal K, \mathbb{R}^{d_y})$ and $C(\mathcal K, \mathbb{R}^{d_y})$ is at least $w^*_{\min} = \max (d_x, d_y)$ for any activation functions. Furthermore, they showed that for some specific activation functions, such as Leaky-ReLU or ReLU+FLOOR, the minimal width can be achieved.

**Summary Of The Review:**

The main result, which states that for any activation functions, the minimum width of a DNN to approximate any functions in $L^p(\mathcal K, \mathbb{R}^{d_y})$ and $C(\mathcal K, \mathbb{R}^{d_y})$ is at least $\max (d_x, d_y)$, is an unexpected and elegant result, which is a non-trivial extension of previous results in this field.

---

> ### Author Response · Authors · 2022-11-11
> **Minor revisions are made according to the comments.**
>
> We thank the reviewer for careful reading and the useful comments and suggestions.
>
> We agree that it is interesting to study the exact minimum width for more activation functions and more architectures used in practice. According to the existing results, such as Johnson (2019), Kidger and Lyons (2020), Park et al. (2021), and the universal lower bound in our manuscript, for activation in practice, such as the sigmoid, tanh and Swish, the exact minimum width for $L^p$-UAP could be bounded by $\max(d_x,d_y)$ and $\max(d_x+2,d_y+1)$. However, obtaining the exact value might be a case-by-case study. We leave it as future works.
>
> For other architecture, there is a quick answer for ResNet: one hidden neuron in each residual block is enough for UAP. The trick is that one can split neurons into lots of residual layers. (See Lin, H.; Jegelka, S. ResNet with One-Neuron Hidden Layers Is a Universal Approximator. In NeurIPS; 2018.)
>
> We have gone through and revised the manuscript according to the comments and suggestions from the reviewers. Hope it solved the concerned issues.

---

> > ### Comment · Reviewer_emi4 · 2022-11-17
> > **Thanks for the reply**
> >
> > Thanks for the detailed reply from the authors. It is interesting to see the comments for Resnet. I will keep the current score.

---

### Official Review · Reviewer_Vm8T · 2022-10-25

**Confidence:** 3
**Clarity, Quality, Novelty And Reproducibility:** See my comments below.
**Correctness:** 4
**Technical Novelty And Significance:** 2
**Empirical Novelty And Significance:** Not applicable
**Recommendation:** 3

**Strength And Weaknesses:**

The presentation needs to be improved. For instance, many terminologies such as C-UAP are not defined.

**Summary Of The Paper:**

The script studies the minimum width of neural networks to achieve universal approximation. Similar topics are also extensively studied in Park et al. 2022.


**Summary Of The Review:**

1. I doubt the significance of the script compared with the existing work Park et al. 2022. I compare the results as follows.

**As for the lower bound:** the script provides a lower bound for arbitrary activation, which extends the results in Park et al which focuses on ReLU.

Park et al. : w_min >= max (d_x +1 , d_y), ReLU

This script: w_min >= max (d_x , d_y), arbitrary activation


**As for the upper bound:** both the script and Park et al. construct neural networks to match the lower bound.

Park et al. : w_min < = max (d_x +2 , d_y+1 ), using general activation

This script: w_min < = max (d_x , d_y), using Leaky-ReLU+ ABS.


For both the upper and lower bound, the script improves the results in Park et al. only by constant 1 or 2. I doubt the significance of such improvement, please clarify the importance if there is any.



2. The most important theoretical result is Theorem 2. However, the relevant analysis (the neuron ODE part) is mainly based on  Li et al. 22 and Duan et al. 22, please clarify the novelty if there is any.


3. The definition of C-UAP and Lp_UAP is missing


++++++++++++++++++ **POST REBUTTAL** ++++++++++++++++++++++

Thanks for the reply from the authors. Regarding my question 1, I am still not convinced about the importance of this result compared with the upper and lower bound of Parker et al.. I will keep my score.

---

> ### Author Response · Authors · 2022-11-11
> **The novelty of this manuscript is clarified.**
>
> We thank the reviewer for the careful reading and useful comments. We have revised the manuscript based on the following responses.
>
> 1. Comparison with the existing works. The results on the minimum width involve lots of studies. Based on previous studies, such as Lu et al. (2017) and Kidger and Lyons (2020), Park et al. (2021) obtained the first exact minimum width results for $L^p$-UAP of ReLU and C-UAP of  ReLU+Step. The obtained minimum width is $\max(d_x+1,d_y)$.
>
>    A natural question is: can we modify the activation functions to achieve the UAP with less number of neurons at each layer (the network width)?
>
>    Our result closed this question. It is beyond a simple improvement on the width bound since our $w_{\min}^*$ is optimal and can not be improved anymore.
>
>    In detail, our manuscript has the following novelties: (1) consider arbitrary activation functions and give a universal minimum width lower bound $w_{\min}^*=\max(d_x,d_y)$; (2) this lower bound can be achieved, for example, the $L^p$-UAP by leaky-ReLU NN; (3) We proposed a novel construction (connected to the neural ODE) which is different from previous coding methods.
>
> 2. The novelty of Theorem 2. Theorem 2 said two things for leaky-ReLU NN: 1) if the width $< w_{\min}$ then UAP is impossible (according to Lemma 1); 2) width = $w_{\min}$ is enough for $L^p$-UAP (according to Lemma 11). The results of Ref [1] Li et al. (2022) and Ref [2] Duan et al. (2022) allow us to prove Lemma 11 quit easily. Note that Ref [1] is for the approximation power of neural ODE/ResNet, and Ref [2] is for approximating some special flow maps. It is not obvious to combine them to study the UAP of FNN. Our observation broadens the implication of the two Refs and provides a nontrivial pipeline to connect the UAP of FNN and neural ODE/ResNet.
>
> 3. Definition of $L^p$-UAP and $C$-UAP. They mean the UAP for the function class $L^p(\mathcal{K},\mathbb{R}^{d_y})$ and $C(\mathcal{K},\mathbb{R}^{d_y})$, with $L^p$ norm and continuous/uniform/sup norm, respectively. We have revised the manuscript to make this notation clear.

---

> ### Author Response · Authors · 2022-11-25
> **Importance of the results.**
>
> We suppose that the reviewer focus on "the upper and lower bound improves only by constant 1 or 2", hence he/she doubts the importance of our manuscript. We have different opinions about this.
>
> First of all, our manuscript is a theoretical study rather than an experimental study. Revealing the theoretical power and limitation of a method (here is the approximation of FNN) is important in its own right.
>
> Secondly, our bounds achieve the critical width. It is beyond a simple improvement since our bound is optimal and can not be improved anymore.
>
> Lastly, we proposed a novel scheme to study the minimum width. Our scheme is different from the encoder-memorizer-encoder scheme proposed by Park et al. In particular, our scheme allows us to prove the leaky-ReLU+ABS NN could achieve the critical width. More importantly, this result could not be obtained (at least not obviously) from the scheme of Park et al.
>
> In a nutshell, our results beyond Park et al.'s not only on the obtained bounds but also the methodology. These give us the confidence to say that our results are important.

---

### Official Review · Reviewer_Mii1 · 2022-10-25

**Confidence:** 3
**Correctness:** 3
**Technical Novelty And Significance:** 3
**Empirical Novelty And Significance:** 3
**Recommendation:** 5

**Clarity, Quality, Novelty And Reproducibility:**

One of the key confusions I have here is about the rationale of proving theorems for the unit cube but claiming that the results hold for any compact domain. Why are the width requirements scale invariant? This claim seems to have been invoked many times but never really explained.

The readability of the paper is quite questionable. For eg. (a) the authors are assuming familiarity with recent results like the ``encoder-memorizer-decoder approach" and hence are not seeming to define any of those terms. Very likely that only a small part of the audience will know of these immediately. This makes the paper feel quite incomplete. Maybe they could have given short descriptions of these in the appendix? (b) The $T_k$ matrices in equation $5$ are quite unclear. What is their definition? Equation 6 seems to give a definition but I am finding it quite hard to parse - where does this come from?

**Strength And Weaknesses:**

The two highlights of their results is that (a) they are able to get a sharp estimate of the width requirement for space of continuous functions mapping between two unit cubes in different dimensions.  And (b) they have a very interesting approach to these width estimation problems via recent results in Neural-ODE.



**Summary Of The Paper:**


This paper builds on certain recent results on Neural-ODE way of approximating functions and gets sharp estimates on the width required for feed-forward nets to approximate continuous and Lp spaces over compact domains.

**Summary Of The Review:**

I think the paper needs some significant improvement in the presentation clarifying the issues raised above and my rating essentially reflects that.

---

> ### Author Response · Authors · 2022-11-11
> **The manuscript has been revised according to the comments and suggestions.**
>
> We thank the reviewer for the careful reading and constructive comments. We have revised the manuscript to make the manuscript more readable.
>
> 1. The difference between the unit cubic and compact domains. The reasons that we only need to consider the unit cubic case are that: 1) any compact domain can be covered by a big cubic, and the functions on the former can be extended to the latter; 2) we can map any cubic to the unit cubic by a linear layer, which is easy for FNN. Although this procedure is standard in math, it's not obvious to most readers. We thank the reviewer pointed out this issue. We have revised the manuscript to include the reasons which will make the manuscript more readable.
> 2. The encoder-memorizer-decoder approach. It includes three parts as listed in Sec 5.3. Figure 3(c) gives an example of the coding procedure. We have revised the manuscript to make this point more clear.
> 3. The map $T_k$ in equation (5). The definition of $T_k^{(i,j)}$ is indeed the equation (6) which is a little bit confusing since it includes many subscripts and superscripts. It comes from the splitting method for solving the ODE (3).  Splitting methods are useful skills in numerical analysis (see McLachlan et al. (2002) for example). The basic idea is to split $v$ as a summation of a few components, $v=v_1+...+v_m$; Then consider the terms $v_i$ one by one; The composition of the split systems, $\dot x(t)=v_i(x(t))$, can approximate the original system $\dot x(t)=v(x(t))$. We have revised this part to make the manuscript easier to follow.

---

### Official Review · Reviewer_VMbe · 2022-10-27

**Confidence:** 3
**Correctness:** 4
**Technical Novelty And Significance:** 3
**Empirical Novelty And Significance:** Not applicable
**Recommendation:** 8

**Clarity, Quality, Novelty And Reproducibility:**

**Clarity, Quality, Novelty:**

The paper is well written and the proofs are easy to understand. I think it is a good contribution that improves the results from past work.

**Strength And Weaknesses:**

**Strengths:**
- They prove matching upper and lower bounds, tightening the bound from Theorem 4 in Park et al. (2021) while also closing the problem of $L^p-$UAP.
- The paper is well written with clear proofs.

**Weaknesses:**
1. I think Table 1 should also include Theorem 4 from Park et al. (2021) which shows that $w_{min} \leq \max(d_x+2, d_y+1)$ for continuous non-poly activations. If I understand correctly, the main contribution of this paper is tightening that bound and proving a corresponding lower bound which Park et al., were lacking. Please correct me if I am wrong.
2. The definition of UOE functions should be stated (atleast intuitively) much earlier in the manuscript as it is mentioned several times.
3. I would rephrase Lemma 11 as a corollary since it follows directly from Li et al., and Duan et al.'s results. That said, it is a nice connection that proves this result quite easily.

**Summary Of The Paper:**

This work tightens the upper bound from Park et al. (2021) while also proving a corresponding lower bound and prove that both $C-$UAP and $L^p-$UAP for functions on compact domains share a common minimum width $w_{min} = \max(d_x, d_u)$. In fact, they use some tools from Neural ODEs to show that the critical width for $L^p-$UAP is achieved by leaky-ReLU networks.

**Summary Of The Review:**

I have a few minor concerns and clarifications as mentioned above. If they are resolved, I would recommend acceptance. I think it is a good contribution to a line of work which has a rich history.

---

> ### Author Response · Authors · 2022-11-11
> **The suggestions are adopted in the revised manuscript.**
>
> We thank the reviewer for the careful reading and helpful comments and suggestions. We have revised the manuscript to adopt the suggestions.
>
> Table 1 is updated to include the results of Park et al. (2021) for continuous non-polynomial activations. A short description for UOE is added at the place where it is first mentioned (page 2). Lemma 11 is changed to Corollary 11, and the cross-references are updated.

---

### Decision · Program_Chairs · 2023-01-20

**Decision:**

Accept: poster

**Justification For Why Not Higher Score:**

It's a extension of the existence theorem, not a newly proposed result.

**Justification For Why Not Lower Score:**

This extension is nontrivial and may of interest to the community.

**Metareview: Summary, Strengths And Weaknesses:**

This work studies the universal approximation by finite width neural networks. It shows with max(d_x, d_y) neurons, a network can approximate any function in L^p(K, R^{d_y}) and C(K, R^{d_y}). Such minimum width can be achieved for some avtivation functions like Leaky ReLU. It extends the results on Park et al. 2022 by considering more general activation functions. This extension is nontrivial and may of interest to the community.  I think the theoretical contribution is enough for the acceptance and I recommend accept.

**Note From Pc:**

if the above contains the word "oral" or "spotlight" please see: "oral" presentation means -> notable-top-5% and "spotlight" means -> notable-top-25%. As stated in our emails, we are disassociating presentation type from AC recommendations